# Fractal Patterns May Illuminate the Success of Next-Token Prediction

**Ibrahim Alabdulmohsin**[*]
Google Deepmind
Zürich, Switzerland
ibomohsin@google.com

**Vinh Q. Tran**
Google Deepmind
New York, USA
vqtran@google.com

**Mostafa Dehghani**
Google Deepmind
Mountain View, USA
dehghani@google.com

## Abstract

We study the fractal structure of language, aiming to provide a precise formalism for quantifying properties that may have been previously suspected but not formally shown. We establish that language is: (1) *self-similar*, exhibiting complexities at all levels of granularity, with no particular characteristic context length, and (2) *long-range dependent* (LRD), with a Hurst parameter of approximately $H = 0.70 \pm 0.09$. Based on these findings, we argue that short-term patterns/dependencies in language, such as in paragraphs, mirror the patterns/dependencies over larger scopes, like entire documents. This may shed some light on how next-token prediction can capture the structure of text across multiple levels of granularity, from words and clauses to broader contexts and intents. In addition, we carry out an extensive analysis across different domains and architectures, showing that fractal parameters are robust. Finally, we demonstrate that the tiny variations in fractal parameters seen across LLMs improve upon perplexity-based bits-per-byte (BPB) in predicting their downstream performance. We hope these findings offer a fresh perspective on language and the mechanisms underlying the success of LLMs.

## 1 Introduction

How does the training objective of predicting the next token in large language models (LLMs) yield remarkable capabilities? Consider, for instance, the two models: Gemini [5] and GPT4 [50]. These models have demonstrated capabilities that extend to quantitative reasoning, summarization, and even coding, which has led some researchers to ponder if there was more to intelligence than "on-the-fly improvisation" [11]. While providing a satisfactory explanation is a difficult endeavor, a possible insight can be drawn from fractals and self-similarity. We elucidate the connection in this work.

**Self-Similarity.** Self-similar processes were introduced by Kolmogorov in 1940 [36]. The notion garnered considerable attention during the late 1960s, thanks to the extensive works of Mandelbrot and his peers [19]. Broadly speaking, an object is called "self-similar" if it is invariant across scales, meaning its statistical or geometric properties stay consistent irrespective of the magnification applied to it (see Figure 1). Nature and geometry furnish us with many such patterns, such as coastlines, snowflakes, the Cantor set and the Kuch curve. Despite the distinction, self-similarity is often discussed in the context of "fractals," another term popularized by Mandelbrot in his seminal book *The Fractal Geometry of Nature* [45]. However, the two concepts are different [26]. See Section 2.

In language, in particular, there have been studies arguing for the presence of a self-similar structure. Nevertheless, due to computational constraints, it was not feasible to holistically model the joint probability distribution of language. As such, linguists often resorted to rudimentary approximations in their arguments, such as by substituting a word with its frequency or length [9], or by focusing on

---

[*]Corresponding author.

38th Conference on Neural Information Processing Systems (NeurIPS 2024).

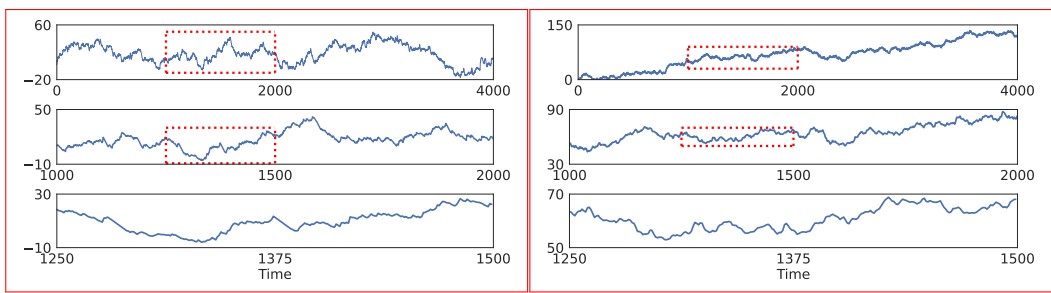

Figure 1: Manifestations of processes across different time scales. A region marked in red corresponds to the magnified plot beneath it. LEFT: The process exhibits self-similarity with rich details at all levels of granularity. It is an integral process $(X_t)_{t \in \mathbb{N}}$ calculated from Wikipedia (see Section 2). RIGHT: Example of a process that is not self-similar, looking smoother at larger time scales.

the recurrence of a specific, predetermined word [49, 3]. These studies fall short of fully capturing the structure of language due to the simplifying assumptions they make, as discussed in Section 4.

Highlighting the self-similar nature of a process can have profound implications. For instance, conventional Poisson models for Ethernet traffic were shown to fail because traffic was self-similar [16, 39, 51, 69]. In such cases, recognizing and quantifying self-similarity had practical applications, such as in the design of buffers [40]. Similarly in language, we argue that self-similarity may offer a fresh perspective on the mechanisms underlying the success of LLMs. Consider the illustrative example shown in Figure 1, where the task is to predict the subsequent measurement in a time series, specifically predicting next tokens in a Wikipedia article (see Section 2 for details). The three plots in Figure 1 (left) represent different manifestations of the same process observed across three distinct time scales. Notably, we observe rich, self-similar details, such as burstiness, in *all* of them. A well-established approach for quantifying self-similarity is the Hölder exponent [66], which we denote by S. In language, we find it to be S $= 0.59 \pm 0.08$, confirming statistical self-similarity.

Why is this important? We hypothesize that since LLMs are trained to predict the future of a self-similar process, they develop proficiency in capturing patterns across multiple levels of granularity for two interconnected reasons. First, self-similarity implies that the patterns at the level of a paragraph are reflective of the patterns seen at the level of a whole text, which is reminiscent of the *recursive* structure of language [53]. Thus, recognizing short-term patterns can aide in learning broader contexts. Second, because language displays intricate patterns at all levels of granularity, it would not be enough to rely only on the immediate context of a sentence to predict the next token. Instead, the model needs to identify patterns at higher levels of granularity; e.g. follow the direction of the argument and the broader intent. It must balance between short- and long-term contexts. Willinger et al. [68] and Altmann et al. [3] argue for self-similarity in language due to this hierarchical nature.

**Long-range dependence.** However, self-similarity alone is not sufficient for a predictive model to exhibit anything resembling "intelligent" behavior. In fact, some self-similar processes, despite their intricate details, remain entirely unpredictable. A quintessential example is the simple Brownian motion, which is a Wiener process with independent increments. Its discrete analog is $B_n = \sum_{i=1}^{n} \varepsilon_i$, where $\varepsilon_i \sim \mathcal{N}(0, \sigma^2)$. Despite possessing rich details at all granularities, a model trained to predict $B_n$ cannot learn anything useful from data since the process itself has *independent* increments.

Thus, for strong capabilities to emerge, the process must have some degree of predictability or dependence as well. One classical metric for quantifying predictability in a stochastic process is the Hurst parameter [31], developed by the hydrologist H. E. Hurst in 1951 while studying the Nile river. It is generally considered to be a robust metric [68], unlike the wavelet estimator [1] and the periodogram method [24] that can be sensitive to errors [54]. As discussed in Section 2.3, we find the Hurst parameter in language to be H $= 0.70 \pm 0.09$. For context, H only takes values in $[0, 1]$. A value H $> 0.5$ implies predictability in the data, while H $= 0.5$ indicates random increments.

While it is compelling that our estimate of H in language lies nearly *midway* between predictability (H $= 1$) and noise (H $= 0.5$), a Hurst parameter of about $0.75$ turns out to occur commonly in nature, including in river discharges, Ethernet traffic, temperatures, precipitation, and tree rings [16, 21, 8]. For agents that learn from data, such as LLMs, this value is also reminiscent of processing-based

theories of curiosity, which suggest that a sweet spot of complexity exists (not too simple, nor too unpredictable) that facilities or accelerates learning [34].

Importantly, predictability and self-similarity *together* imply long-range dependence (LRD). This follows from the definition of self-similarity, where the patterns at small scales mirror those at larger scales so, for example, the correlations established at micro levels are also pertinent at macro levels. LRD is arguably crucial for enhancing the functionality of predictive models because processes with only short-range dependence could be forecasted (somewhat trivially) with lookup tables that provide the likelihood of transitions over brief sequences. By contrast, this is not possible in LRD processes whose contexts extend indefinitely into the past.

**Information Theoretic Complexity.**   To define fractal parameters, we follow recent works such as [28, 22, 41, 47, 25] in adopting an *information-theoretic* characterization of the complexity in language using minimal-length codes or surprise. This corresponds to an intrinsic, *irreducible* description of language and the minimum compute overhead to comprehend/decode it [22], which also correlates well with actual reading times [28, 41]. In this context, self-similarity means that the intrinsic complexity or surprise in language (measured in bits) cannot be smoothed out, even as we look into broader narratives. That is, surprising paragraphs will follow predictable paragraphs, in a manner that is statistically similar to how surprising sentences follow predictable sentences.

**Analysis.**   How robust are these findings? To answer this question, we carry out an extensive empirical analysis across various model architectures and scales, ranging from 1B to over 500B parameters. We find that fractal parameters are quite robust to the choice of the architecture.

However, there exists *tiny* variations across LLMs. Interestingly, we demonstrate that from a practical standpoint, these differences help in predicting downstream performance in LLMs compared to using perplexity-based metrics alone, such as bits-per-byte (BPB). Specifically, we introduce a new metric and show that using it to predict downstream performance can increase the adjusted $R^2$ from approximately $0.65$ when using solely BPB, to over $0.86$ with the new metric[2].

**Statement of Contribution.** In summary, we:

1. highlight how the fractal structure of language can offer a new perspective on the capabilities of LLMs, and provide a formalism to quantify properties, such as long-range dependence.

2. establish that language is self-similar and long-range dependent. We provide concrete estimates in language of the three parameters: the self-similarity (Hölder) exponent, the Hurst parameter, and the fractal dimension. We also estimate the related Joseph exponent.

3. carry out a comparative study across different model architectures and scales, and different domains, such as ArXiv and GitHub, demonstrating that fractal parameters are robust.

4. connect fractal patterns with learning. Notably, we show that a "median" Hurst exponent improves upon perplexity-based bits-per-byte (BPB) in predicting downstream performance.

## 2   Fractal Structure of Language

### 2.1   Preliminaries

Suppose we have a discrete-time, stationary stochastic process $(x_t)_{t \in \mathbb{N}}$, with $\mathbb{E}[x_t] = 0$ and $\mathbb{E}[x_t^2] = 1$. We will refer to $(x_t)_{t \in \mathbb{N}}$ as the *increment process* to distinguish it from the *integral process* $(X_t)_{t \in \mathbb{N}}$ defined by $X_t = \sum_{k=0}^{t} x_k$. While $(x_t)_{t \in \mathbb{N}}$ and $(X_t)_{t \in \mathbb{N}}$ are merely different representations of the same data, it is useful to keep both representations in mind. For example, self-similarity is typically studied in the context of integral processes whereas LRD is defined on increment processes.

In the literature, it is not uncommon to mistakenly equate parameters that are generally different. For example, the Hurst parameter H has had many definitions in the past that were not equivalent, and Mandelbrot himself cautioned against this [44]. The reason behind this is because different parameters can agree in the idealized fractional Brownian motion, leading some researchers to equate them in general [66]. We will keep the self-similarity exponent S and H separate in our discussion.

---

[2]We release the code for calculating fractal parameters at: https://github.com/google-research/google-research/tree/master/fractals_language

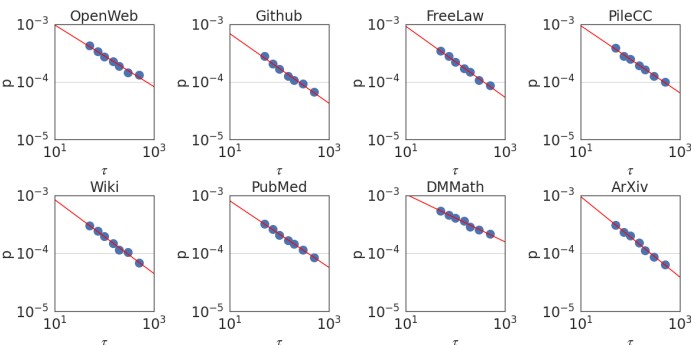

Figure 2: Peak probability $p_\epsilon(\tau)$ is plotted against the granularity level $\tau$ (see Section 2.2). We observe power laws $p_\epsilon(\tau) \sim \tau^{-S}$, indicating self-similarity, with a median exponent of S $= 0.59 \pm 0.08$.

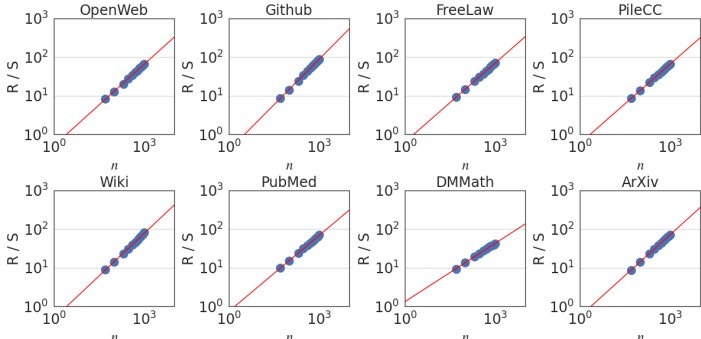

Figure 3: Rescaled range $R(n)/S(n)$ is plotted against the number of normalized bits $n$. We observe a power law $R(n)/S(n) \sim n^{\mathrm{H}}$ in all domains. When aggregating all datasets, H $= 0.70 \pm 0.09$.

**Experimental Setup.** In order to establish self-similarity and LRD in language, we convert texts into sequences of bits using a large language model (LLM). Specifically, we use PaLM2-L (Unicorn) [6] to calculate the probability of the next token $w_t$ conditioned on its entire prefix $w_{[t-1]} = (w_0, w_1, \ldots, w_{t-1})$. As discussed in Section 1, this captures its intrinsic, irreducible description [22]. By the chain rule [15], the corresponding number of bits assigned to $w_t$ is $z_t = -\log p(w_t|w_{[t-1]})$. Unlike in prior works, which rely on simplifications such as by substituting a word with its length [9] or by focusing on the recurrence of a single word [49, 3], we use the LLM to approximate the full joint distribution of language since LLMs are known to produce calibrated probability scores at the token level [33]. We carry out these calculations for prefixes of up to 2048 tokens ($\approx 8$ pages of text). With a suitable normalization, such as bits-per-byte (BPB), one obtains a standardized description of text, consistent across tokenizers. BPB is widely used as a tokenizer-agnostic metric to compare LM modeling performance, e.g. for The Pile [23].

Besides PaLM2, we also experiment and report on various model sizes of PaLM [12] and decoder-only T5 [55]. Namely, we report results for models: PaLM2 XXS (Gecko), XS (Otter), S (Bison), M, and L (Unicorn); PaLM 8B, 62B, 540B; and decoder-only T5.1.1 at Base (110M), Large (341M), XL (1.2B), and XXL (5B) sizes. For PaLM and PaLM2, we use the checkpoints pretrained in Chowdhery et al. [12] and Anil et al. [6]. All T5.1.1 decoder baselines, on the other hand, are trained with a casual language modeling objective for 262B tokens of C4 [55]. All experiments are executed on Tensor Processing Units (TPUs). More details on how we train T5.1.1 baselines are in Appendix A.

Once $z_t$ is computed for a document, we follow standard definitions in constructing the increment process $(x_t)_{t \in \mathbb{N}}$ by normalizing $z_t$ to have a zero-mean and unit variance. Intuitively, fractal parameters are intended to measure a fundamental property of the process (e.g. LRD) that should not be affected by scale, hence the normalization. The integral process $(X_t)_{t \in \mathbb{N}}$ is calculated based on $(x_t)_{t \in \mathbb{N}}$, as described earlier and depicted in Figure 1 (top). Normalizing bits (to have zero mean and unit variance) models language as a random walk. It is a standard approach used extensively in the literature in various contexts, such as in DNA sequences [52, 57, 48, 35, 59].

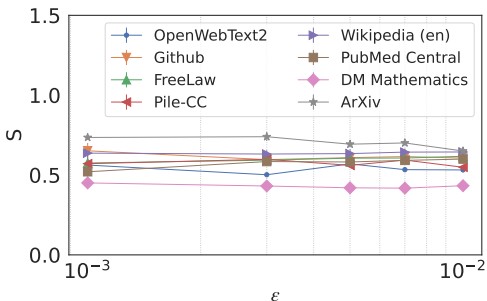 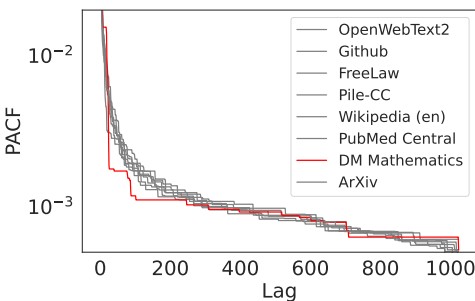

Figure 4: LEFT: Estimates of the self-similarity exponent S are generally robust to the choice of $\epsilon$. RIGHT: The partial auto-correlation function calculated across domains. DM Mathematics has a much shorter dependence compared to the rest of the domains, in agreement with its Hurst parameter.

For analysis, we use The Pile validation split [23], consisting of 22 subdomains such as Wikipedia and GitHub. We restrict analysis to sufficiently-long documents of length $> 4K$ tokens and use the first 2K tokens only, to sidestep potential effects of the finite length of documents and the model context. To mitigate noise, only domains with $> 1K$ documents are compared; we report results for them separately and their median. We use bootstrapping [17] to estimate the error margin.

**Notation.** We write $f(x) \sim x^c$ if $f(x) = x^c L(x)$ for some function $L$ that satisfies $L(tx)/L(x) \to 1$ as $x \to \infty$ for all $t > 0$. Examples of slowly varying functions are constants $L(x) = c$ and $L(x) = \log x$. When $f(x) \sim x^c$, we abuse terminology slightly by referring to $f(x)$ as a power law.

## 2.2 Self-similarity exponent — Scale invariance

An integral process is said to be self-similar if it exhibits *statistical* self-similarity. More precisely, $(X_t)_{t\in\mathbb{N}}$ is self-similar if $(X_{\tau t})_{t\in\mathbb{N}}$ is distributionally equivalent to $(\tau^S X_t)_{t\in\mathbb{N}}$ for some exponent S. Thus, scaling of time is equivalent to an appropriate scaling of space. We will refer to $\tau$ as the *granularity level* and to the exponent S as the self-similarity or Hölder exponent [66]. Many time series in nature exhibit self-similar structures, such as human blood pressure and heart rate [27].

One approach for calculating S is as follows. Fix $\epsilon \ll 1$ and denote the $\tau$-increments by $(X_{t+\tau} - X_t)_{t\in\mathbb{N}}$. These would correspond, for instance, to the number of bits used for clauses, sentences, paragraphs and longer texts as $\tau$ increases. In terms of the increment process $(x_t)_{t\in\mathbb{N}}$, this corresponds to aggregating increments into "bursts". Let $p_\epsilon(\tau)$ be the probability mass of the event $\{|X_{t+\tau}-X_t| \leq \epsilon\}_{t\in\mathbb{N}}$. Then, S can be estimated by fitting a power law relation $p_\epsilon(\tau) \sim \tau^{-S}$ [66]. Generally, S is robust to the choice of $\epsilon \in [10^{-3}, 10^{-2}]$ as shown in Figure 4 (left) so we fix it to $\epsilon = 5 \times 10^{-3}$.

Figure 2 plots the probability $p_\epsilon(\tau)$ against $\tau$ using PaLM2-L. We indeed observe a power law relation over at least two orders of magnitude; i.e. linear in a log-log scale, with a median self-similarity exponent of $S = 0.59 \pm 0.08$. Section 3 shows that the median S is robust to the choice of the LLM.

## 2.3 Hurst parameter — Long-range dependence

The Hurst parameter $H \in [0, 1]$ quantifies the degree of predictability or dependence over time [31]. It is calculated using the so-called rescaled-range (R/S) analysis. Let $(x_t)_{t\in\mathbb{N}}$ be an increment process. For each $n \in \mathbb{N}$, write $y_t = x_t - \frac{1}{t}\sum_{k=0}^{t} x_k$ and $Y_t = \sum_{k=0}^{t} y_t$. The range and scale are defined, respectively, as $R(n) = \max_{t\leq n} Y_t - \min_{t\leq n} Y_t$ and $S(n) = \sigma(\{x_k\}_{k\leq n})$, where $\sigma$ is the standard deviation. Then, the Hurst parameter H is estimated by fitting a power law relation $R(n)/S(n) \sim n^H$. As stated earlier, for completely random processes, such as a simple Brownian motion, it can be shown that $H = 1/2$. In addition, $H > 1/2$ implies dependence over time [16, 68, 8].

Writing $\rho_n = \mathbb{E}[(x_{t+n}x_t)]$ for the autocovariance function of the increment process $(x_t)_{t\in\mathbb{N}}$, the Hurst parameter satisfies $H = 1 - \beta/2$ when $\rho_n \sim n^{-\beta}$ as $n \to \infty$ [26, 16]. Since in self-similar processes, $H > 1/2$ implies long-range dependence (LRD), LRD is equivalent to the condition that the autocovariances are not summable. In terms of the integral process, it can be shown that [58]: $\lim_{n\to\infty} \frac{\text{Var}(X_n)}{n} = 1 + 2\sum_{i=1}^{\infty} \rho_i$. Hence, if $H < 1/2$, the auto-covariances are summable and

|   | OpenWeb | GitHub | FreeLaw | PileCC | Wiki | PubMed | Math | ArXiv |
|---|---------|--------|---------|--------|------|--------|------|-------|
| S | $0.53 \pm .05$ | $0.60 \pm .05$ | $0.61 \pm .05$ | $0.56 \pm .03$ | $0.62 \pm .02$ | $0.60 \pm .07$ | $0.42 \pm .03$ | $0.70 \pm .03$ |
| H | $0.68 \pm .01$ | $0.79 \pm .01$ | $0.68 \pm .00$ | $0.70 \pm .00$ | $0.74 \pm .01$ | $0.65 \pm .00$ | $0.50 \pm .01$ | $0.72 \pm .01$ |
| J | $0.46 \pm .01$ | $0.49 \pm .00$ | $0.49 \pm .00$ | $0.50 \pm .00$ | $0.52 \pm .00$ | $0.44 \pm .00$ | $0.28 \pm .00$ | $0.49 \pm .00$ |

Table 1: A comparison of the fractal parameters across 8 different domains with $> 1000$ documents each in The Pile benchmark (see Section 2.1 for selection criteria). DM-Mathematics is markedly different because each document consists of questions, with no LRD.

$\text{Var}(X_n)$ grows, at most, linearly fast on $n$. On the other hand, if the process has LRD, $\text{Var}(X_n)$ grows superlinearly on $n$. In particular, using the Euler-Maclaurin summation formula [7, 2], one obtains $\text{Var}(X_n) \sim n^{2H}$ if $H > 1/2$. Figure 3 plots the rescaled range $R(n)/S(n)$ against $n$. We observe a power law relation with a median Hurst parameter of $H = 0.70 \pm 0.09$.

### 2.4 Fractal dimension — Complexity at all levels

Broadly speaking, the fractal dimension of an object describes its *local* complexity. For a geometric object $Z$, such as the Koch curve, let $\tau$ be a chosen scale (e.g. a short ruler for measuring lengths or a small square for areas). Let $N(\tau)$ be the minimum number of objects of scale $\tau$ that cover $Z$; i.e. contain it entirely. Then, the fractal dimension of $Z$, also called its Hausdorff dimension, is: $D = -\lim_{\tau \to 0} \left\{ \frac{\log N(\tau)}{\log \tau} \right\}$ [54]. For example, a line has a fractal dimension 1, in agreement with its topological dimension, because $N(\tau) = C/\tau$ for some constant $C > 0$.

By convention, an object is referred to as "fractal" if D is different from its topological dimension. For example, the fractal dimension of the Koch curve is about 1.26 when its topological dimension is 1. Fractals explain some puzzling observations, such as why estimates of the length of the coast of Britain varied significantly from one study to another, because lengths in fractals are scale-sensitive. Mandelbrot estimated the fractal dimension of the coast of Britain to be 1.25 [43].

The definition above for the fractal dimension D applies to geometric shapes, but an analogous definition has been introduced for stochastic processes. Let $(x_t)_{t \in \mathbb{R}}$ be a stationary process with autocovariance $\rho_n$. Then, its fractal dimension D is determined according to the local behavior of $\rho_n$ at the vicinity of $n = 0$, by first normalizing $(x_t)_{t \in \mathbb{R}}$ to have a zero-mean and a unit variance, and modeling $\rho_n$ using a power law $\rho_n \sim 1 - n^\alpha$ as $n \to 0^+$, for $\alpha \in (0, 2]$. Then, the fractal dimension $D \in [1, 2]$ of $(x_t)_{t \in \mathbb{R}}$ is defined by $D = 2 - \alpha/2$ [26]. It can be shown that $D = 2 - S$ [26]. For language, this gives a median fractal dimension of $D = 1.41 \pm 0.08$.

### 2.5 Joseph effect — Burstiness

Finally, we examine another related parameter that is commonly studied in self-similar processes. The motivation behind it comes from the fact that in processes with LRD, one often observes *burstiness* as shown in Figure 1; i.e. clusters over time in which the process fully resides on one side of the mean, before switching to the other. This is quite unlike random noise, for instance, where measurements are evenly distributed on both sides of the mean. The effect is often referred to as the Joseph effect, named after the biblical story of the seven fat years and seven lean years [68, 46, 66].

A common way to quantify the Joseph effect for integral processes $(X_t)_{t \in \mathbb{N}}$ is as follows [66]. First, let $\sigma_\tau$ be the standard deviation of the $\tau$-increments $X_{t+\tau} - X_t$. Then, fit a power law relation $\sigma_\tau \sim \tau^J$. The exponent J here is called the Joseph exponent. In an idealized fractional Brownian motion, both J and the self-similarity exponent S coincide. Figure 5 provides the detailed empirical results. Overall, we find that $J = 0.49 \pm 0.08$.

## 3 Analysis

**Comparative Analysis.** Table 1 compares fractal parameters across different domains, such as ArXiv, Github and Wikipedia. In general, most domains share similar self-similarity and Hurst exponents with a few exceptions. The first notable exception is DM-Mathematics, which has a Hurst parameter of about 0.5, indicating a lack of LRD. Upon closer inspection, however, a value of $H = 0.5$

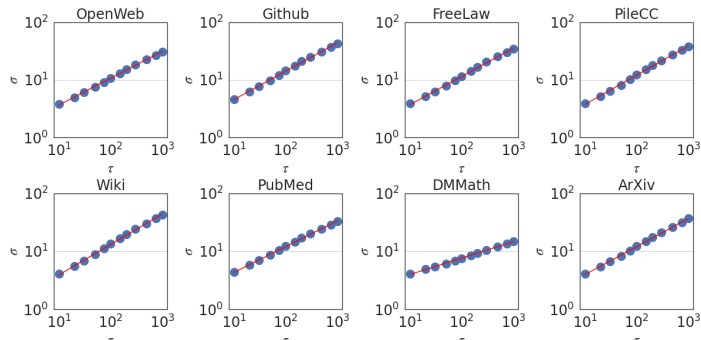

Figure 5: The standard deviation $\sigma$ of the $\tau$-increments $X_{t+\tau} - X_t$ is plotted against the scale $\tau$. We, again, observe another power law relation $\sigma \sim \tau^{\mathrm{J}}$, with a Joseph exponent $\mathrm{J} = 0.49 \pm 0.08$.

| | T5-Decoder | | | | PaLM | | | | PaLM2 | | |
|---|---|---|---|---|---|---|---|---|---|---|---|
| 110M | 340M | 1B | 5B | 8B | 62B | 540B | XXS | XS | S | M | L |
| | | | | | Self-similarity exponent S | | | | | | |
| $.58^{\pm.06}$ | $.60^{\pm.06}$ | $.60^{\pm.05}$ | $.58^{\pm.08}$ | $.60^{\pm.07}$ | $.62^{\pm.08}$ | $.64^{\pm.08}$ | $.59^{\pm.06}$ | $.57^{\pm.08}$ | $.56^{\pm.05}$ | $.59^{\pm.07}$ | $.60^{\pm.08}$ |
| | | | | | Hurst exponent H | | | | | | |
| $.64^{\pm.08}$ | $.64^{\pm.08}$ | $.64^{\pm.09}$ | $.64^{\pm.08}$ | $.66^{\pm.07}$ | $.68^{\pm.07}$ | $.68^{\pm.07}$ | $.66^{\pm.07}$ | $.66^{\pm.07}$ | $.67^{\pm.08}$ | $.68^{\pm.09}$ | $.69^{\pm.09}$ |
| | | | | | Joseph exponent J | | | | | | |
| $.44^{\pm.06}$ | $.44^{\pm.06}$ | $.44^{\pm.06}$ | $.44^{\pm.06}$ | $.47^{\pm.06}$ | $.47^{\pm.06}$ | $.48^{\pm.06}$ | $.47^{\pm.06}$ | $.47^{\pm.06}$ | $.48^{\pm.07}$ | $.48^{\pm.07}$ | $.49^{\pm.08}$ |

Table 2: A comparison of the estimated median fractal parameters by various LLMs over the entire Pile validation split. Estimates are generally robust to the choice of the LLM, but the tiny variations in median H reflect improvements in the model quality. See Section 3.

is not surprising for DM-Mathematics because its documents consist of independent mathematical questions as shown in Figure 6. In Figure 4 (right), we plot the partial autocorrelation function for each of the 8 domains against time lag (context length). Indeed, we see that DM-Mathematics shows markedly less dependence compared to the other domains. The second notable observation is the relatively larger value of $\mathrm{H} = 0.79$ in GitHub, indicating more structure in code. This is in agreement with earlier findings by Kokol and Podgorelec [35] who estimated LRD in computer languages to be greater than in natural language. In Table 2, we compare the three fractal parameters S, H and J using different families of LLM and different model sizes. Overall, we observe that the parameters are generally robust to the choice of the architecture.

**Downstream Performance.** By definition, fractal parameters are calculated on the sequence of negative log-probability scores after normalizing them to zero-mean and unit variance. Hence, they may offer an assessment of downstream performance that improves upon using a perplexity-based metric like bits-per-byte (BPB) alone. To test this hypothesis, we evaluate the 12 models in Table 2 on challenging downstream zero- and few-shot benchmarks focusing on language understanding and reasoning. We include results for 0-shot (0S) and 3-shot (3S) evaluation for BIG-Bench Hard tasks [63, 64] reporting both direct and chain-of-thought (CoT) prompting results following Chung et al. [13]. In addition we report 0-shot and 5-shot (5S) MMLU [30], and 8-shot (8S) GSM8K [14] with CoT. Raw accuracy is reported for all tasks. BBH and MMLU scores are averaged across all 21 tasks and 57 subjects, respectively. These benchmarks are quite diverse and include tasks such as logical deduction, arithmetic, translation error detection, disambiguation, as well as general knowledge (e.g. history, computer science, law, sports, movies, etc). All prompt templates for our evaluation are taken from Chung et al. [13], Longpre et al. [42], which we refer the reader to for more details. We prompt all models using a 2048 context length. See Table 8 of Appendix C for the full results.

The first (surprising) observation is that the median Hurst parameter is itself strongly correlated with the BPB scores with an absolute Pearson correlation coefficient of 0.83, even though the Hurst exponent is calculated after normalizing all token losses to zero-mean and unit variance! Informally,

```
Document I:  What is the square root of 211269 to the nearest integer?  460.  What is the square root
of 645374 to the nearest integer?  803...
Document II: Suppose 5*l = r - 35, -2*r + 5*l - 15 = -70.  Is r a multiple of 4?  True.  Suppose 2*l +
11 - 1 = 0.  Does 15 divide (-2)/l - 118/(-5)?  False...
```

Figure 6: Two examples of documents from the DM-Mathematics subset of The Pile benchmark [23]. Each document comprises of multiple independent questions. The lack of LRD in this data is reflected in its Hurst parameter of $H = 0.50 \pm 0.01$

| Benchmark | BPB | H | $H_B$ | 2K | 4K | 8K |
|---|---|---|---|---|---|---|
| 0S BBH Direct | 0.785 | 0.841 | **0.883** | 1.81 | 1.68 | 1.76 |
| 0S MMLU | 0.653 | **0.831** | **0.825** | 25.73 | 26.04 | 25.81 |
| 0S BBH+MMLU | 0.685 | **0.849** | **0.852** | 13.39 | 13.49 | 13.42 |
| 3S BBH Direct | 0.767 | 0.895 | **0.926** | 21.35 | 24.76 | 23.14 |
| 3S BBH CoT | 0.881 | 0.892 | **0.979** | 16.87 | 12.21 | 7.14 |
| 5S MMLU | 0.660 | **0.853** | 0.832 | 26.57 | 26.69 | 27.07 |
| 8S GSM8K CoT | 0.654 | **0.867** | 0.851 | 1.06 | 1.21 | 1.74 |
| FS BBH+MMLU+GSM8K | 0.717 | **0.890** | **0.891** | 15.58 | 15.46 | 14.65 |

Table 3: MIDDLE three columns show the adjusted $R^2$: the proportion of variation in downstream performance (row) predictable by a linear function of the input (column). Median Hurst (H) and (especially) the combined metric $H_B$ predict downstream performance better than BPB alone. S and J do not give any improvement (see Appendix C). RIGHT: the downstream performance for three decoder-only T5.1.1. models pretrained on 100B tokens with 2K, 4K, or 8K context lengths.

this implies that second-order statistics on the sequence of token losses of a particular model can predict its mean! Self-similarity exponent, by contrast, has an absolute correlation of 0.23 with BPB.

Figure 7 displays downstream performance against both the median Hurst exponent and the median BPB score, where median values are calculated on the 8 domains in The Pile benchmark listed in Table 1. In general, both the BPB score and the median Hurst are good predictors of downstream performance. However, we observe that improvements in BPB alone without impacting the median Hurst exponent do not directly translate into improvements downstream. This is verified quantitatively in Table 3 (middle), which reports the adjusted $R^2$ values – the proportion of variance in each downstream metric that can be predicted using BPB, H, or by combining them together into $H_B = 1/\text{BPB} + H$, with BPB replaced with its reciprocal so that higher values are better. We observe that $H_B$ yields indeed a stronger predictor of downstream performance. Hence, while H and BPB are correlated, combining them yields a *better* predictor, so each of H and BPB conveys useful information not captured by the other metric. See Appendix C for similar analysis using the exponents S and J.

**Context Length at Training Time (*Negative Result*).**    Finally, we present a negative result. Self-similarity and LRD point to an intriguing possibility: the importance of *training* the model with extensive contexts in order to capture the fractal-nature of language, which may elevate the model's capabilities regardless of the context length needed during inference. To test this hypothesis, we pretrain three decoder-only T5.1.1 models with 1B parameters on SlimPajama-627B [62] for up to 100B tokens using three context lengths: 2K, 4K and 8K, all observing the same number of tokens per batch. We use SlimPajama-627B instead of C4 because most documents in C4 are short ($\approx 94\%$ of them are $< 2K$ tokens in length). Refer to Appendix A for details. These models are, then, evaluated on the same downstream benchmarks listed in Figure 7. As shown in Table 3 (right) however, we do not observe any improvements in performance with context length in this particular setup.

## 4   Related Works and Directions for Future Research

The statistical attributes of human language have long piqued scholarly curiosity. One example is Zipf's law, which Shannon leveraged to estimate the entropy of English to be around 1 bit per letter [60], but his calculation did not consider second-order statistics. More recently, Eftekhari [18] proposed a refinement to Zipf's law, suggesting its application to letters rather than words. Another related result is Heap's law, which states that the number of unique words is a power law function

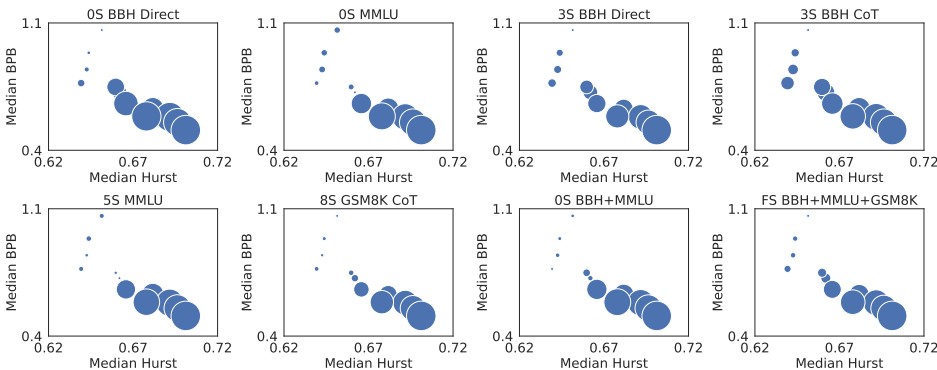

Figure 7: Downstream metric, indicated by bubble size where larger is better, is plotted vs. the median Hurst and the median BPB for all 12 language models.

of the document's length [29]. However, both Zipf's and Heap's laws are invariant to the semantic ordering of text, so they do not capture important aspects, such as long-range dependence (LRD) [49].

In terms of self-similarity in language, the Menzerath-Altmann law stipulates a self-similar behavior in the following sense: when the size of a language construct increases, the size of its constituents decreases, and this happens at all scales [49, 4]. In Ausloos [9], the authors model texts as a time series by replacing a word with its length. After that, they study the fractal behavior of language. However, as mentioned in [22], replacing a word with its length is invalid because it is not translation-independent (i.e. one could map every word to an arbitrary token, including tokens of equal length). In our work, we model language as a series of bits calculated from conditional entropies, reflecting the intrinsic structure of the language itself, inspired by findings in linguistics such as [28, 22, 41]. The existence of self-similarity in language is attributed to its hierarchical nature [68, 3], such as duality of patterning [37].

In Najafi and Darooneh [49], the authors define a fractal dimension for each word. Informally, they examine the recurrence of a single, predetermined word as a binary series, similar to the approach used in Altmann et al. [3]. However, this only applies to individual words and cannot model higher-level clauses. For instance, it does not distinguish between "time" in the phrase "once upon a time" and "time" in "space and time." Kokol and Podgorelec [35] estimate LRD in natural language, and suggest that its LRD is close to that of pure noise! They conjecture this was due to their use of ASCII encoding. In computer languages, they observe LRD and suggest it is because they are formal.

Besides the above concerns in prior studies that examined the self-similar structure in language, another concern is that they sometimes give extremely large values of the fractal dimension, sometimes exceeding 10 [4]! Such values are difficult to interpret because the fractal dimension D should fall in $D \in [1, 2]$ for time series. We do not observe such issues in our analysis. In our case, $D = 1.41 \pm 0.08$.

**Limitations and Future Research.** Our analysis is currently limited to the English language so it may not apply to other languages that differ significantly. For instance, some languages such as Pirahã (spoken in the Amazon) do not have a recursive structure like most languages do [20]. We also do not model the semantic or lexical form of language. While our information-theoretic approach is well-founded and captures the intrinsic complexity of language, it does not account for the semantic nuances that contribute to meaning. Thirdly, self-similarity may explain why parameter sharing, such as in ALBERT [38], can be successful but exploiting self-similarity more directly in LLMs could lead to further optimizations. Exploring these aspects are promising directions for future research.

## 5   Concluding Remarks

In this work, we highlight intriguing insights into the underlying fractal structure of language and how it may be interconnected with the remarkable capabilities of LLMs. Our formalism quantifies properties of language that may have been suspected, but not previously formally shown. In particular, the need in LLMs to balance between short- and long-term contexts is reflected in the self-similar

structure of language, while long-range dependence is quantifiable using the Hurst parameter. For instance, the absence of LRD in DM-Mathematics is reflected in its Hurst parameter of $H \approx 0.5$. Interestingly, the estimated median Hurst value of $H = 0.70 \pm 0.09$ in language reflects an intriguing balance between predictability and noise that is similar to many other phenomena, and combining both $H$ with BPB together yields a stronger predictor of downstream performance. We carry out an extensive comparative analysis across different domains and model architectures, revealing that fractal parameters are generally robust. We hope that future research can further probe into these fractal properties, unearthing deeper understandings of the relation between intelligence and language.

## Acknowledgement

The authors would like to thank Justin Gilmer and Olivier Bousquet for their feedback on earlier drafts of this manuscript, and both Google Deepmind and Google Research teams at large for the insightful discussions and providing a supportive research environment.

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

# A  Experiment Details

All of our experiments are conducted in JAX/Flax [10] using the open source T5X framework [56].

T5 baselines in Table 2 and 3 are pretrained from scratch using the open source T5.1.1 decoder-only architecture from the T5X library.[3] We pretrain using a causal language modeling objective over the C4 corpus with the default T5 vocabulary as per Raffel et al. [55]. Training is done for 500k steps with a sequence length of 1024 and batch size of 512, resulting in a total of 262B tokens seen during pretraining. We optimize our model with the Adafactor [61] optimizer with an inverse square root learning rate schedule, 1k warmup steps, and an initial learning rate of 1e-2. Models are trained using 256 TPUv5e chips [32].

T5 context length ablation experiments in Table 3 are trained with the same pretraining objective but over the SlimPajama-627B corpus [62] and using a modified version of the T5 vocabulary that preserves whitespace and introduces byte-fallback for out of vocabulary tokens. This is similar to Chowdhery et al. [12], but preserving the original T5 vocabulary. Models with sequence lengths 2048, 4096, 8192 are trained with batch sizes of 512, 256, and 128 respectively to preserve the number of tokens seen per batch and overall training steps. We train all models for 100k steps, using the same learning rate schedule described above. Hence, all models observe 100B tokens.

---

[3]https://github.com/google-research/t5x/tree/main/t5x/examples/decoder_only/models

# B  Full Results

In this section, we provide the full list of parameters calculated for each combination of LLM and domain. We use bootstrapping [17] to estimate the error margin.

| Model | OpenWebText2 | Github | FreeLaw | Pile-CC | Wikipedia | PubMed | Mathematics | ArXiv |
|---|---|---|---|---|---|---|---|---|
| T5-Decoder-110M | 2.89 | 1.82 | 2.45 | 2.88 | 2.80 | 2.36 | 2.28 | 2.70 |
| T5-Decoder-340M | 2.60 | 1.56 | 2.14 | 2.62 | 2.52 | 2.08 | 2.10 | 2.42 |
| T5-Decoder-1B | 2.38 | 1.37 | 1.91 | 2.41 | 2.29 | 1.88 | 2.00 | 2.19 |
| T5-Decoder-5B | 2.19 | 1.22 | 1.73 | 2.25 | 2.11 | 1.73 | 1.91 | 2.01 |
| PaLM1-8B | 2.26 | 0.79 | 1.66 | 2.36 | 2.08 | 1.89 | 1.40 | 2.08 |
| PaLM1-62B | 2.02 | 0.62 | 1.44 | 2.14 | 1.80 | 1.68 | 1.30 | 1.83 |
| PaLM1-540B | 1.88 | 0.54 | 1.33 | 2.01 | 1.58 | 1.57 | 1.25 | 1.68 |
| PaLM2-XXS | 2.37 | 0.87 | 1.77 | 2.46 | 2.17 | 1.96 | 1.38 | 1.96 |
| PaLM2-XS | 2.12 | 0.73 | 1.53 | 2.22 | 1.92 | 1.72 | 1.27 | 1.72 |
| PaLM2-S | 1.95 | 0.60 | 1.37 | 2.06 | 1.71 | 1.57 | 1.19 | 1.55 |
| PaLM2-M | 1.88 | 0.56 | 1.31 | 1.99 | 1.59 | 1.51 | 1.12 | 1.48 |
| PaLM2-L | 1.75 | 0.46 | 1.23 | 1.88 | 1.22 | 1.43 | 1.08 | 1.36 |

Table 4: Log-perplexity (NLL) scores evaluated on the first 2048 tokens, after trimming the first 100 tokens, of documents belonging to each of the shown domains. Only documents with a minimum length of 4K tokens are used.

| Model | OpenWebText2 | Github | FreeLaw | Pile-CC | Wikipedia | PubMed | Mathematics | ArXiv |
|---|---|---|---|---|---|---|---|---|
| T5-Decoder-110M | $0.58 \pm 0.04$ | $0.67 \pm 0.03$ | $0.51 \pm 0.02$ | $0.54 \pm 0.07$ | $0.59 \pm 0.04$ | $0.59 \pm 0.03$ | $0.51 \pm 0.04$ | $0.58 \pm 0.05$ |
| T5-Decoder-340M | $0.52 \pm 0.03$ | $0.59 \pm 0.05$ | $0.63 \pm 0.04$ | $0.58 \pm 0.04$ | $0.61 \pm 0.03$ | $0.61 \pm 0.03$ | $0.48 \pm 0.04$ | $0.61 \pm 0.05$ |
| T5-Decoder-1B | $0.54 \pm 0.01$ | $0.66 \pm 0.11$ | $0.61 \pm 0.06$ | $0.57 \pm 0.06$ | $0.59 \pm 0.05$ | $0.60 \pm 0.02$ | $0.50 \pm 0.03$ | $0.63 \pm 0.02$ |
| T5-Decoder-5B | $0.51 \pm 0.04$ | $0.70 \pm 0.04$ | $0.60 \pm 0.04$ | $0.58 \pm 0.02$ | $0.58 \pm 0.03$ | $0.57 \pm 0.02$ | $0.45 \pm 0.02$ | $0.67 \pm 0.05$ |
| PaLM1-8B | $0.56 \pm 0.03$ | $0.67 \pm 0.05$ | $0.63 \pm 0.05$ | $0.58 \pm 0.01$ | $0.55 \pm 0.04$ | $0.62 \pm 0.03$ | $0.50 \pm 0.03$ | $0.68 \pm 0.07$ |
| PaLM1-62B | $0.49 \pm 0.03$ | $0.65 \pm 0.09$ | $0.63 \pm 0.09$ | $0.57 \pm 0.03$ | $0.63 \pm 0.05$ | $0.61 \pm 0.04$ | $0.48 \pm 0.05$ | $0.68 \pm 0.03$ |
| PaLM1-540B | $0.51 \pm 0.04$ | $0.68 \pm 0.09$ | $0.64 \pm 0.05$ | $0.58 \pm 0.04$ | $0.67 \pm 0.03$ | $0.64 \pm 0.08$ | $0.48 \pm 0.03$ | $0.65 \pm 0.04$ |
| PaLM2-XXS | $0.53 \pm 0.02$ | $0.61 \pm 0.05$ | $0.58 \pm 0.04$ | $0.60 \pm 0.04$ | $0.57 \pm 0.05$ | $0.61 \pm 0.03$ | $0.52 \pm 0.02$ | $0.70 \pm 0.04$ |
| PaLM2-XS | $0.54 \pm 0.04$ | $0.57 \pm 0.06$ | $0.58 \pm 0.03$ | $0.56 \pm 0.04$ | $0.60 \pm 0.04$ | $0.57 \pm 0.06$ | $0.45 \pm 0.02$ | $0.73 \pm 0.06$ |
| PaLM2-S | $0.55 \pm 0.02$ | $0.55 \pm 0.15$ | $0.59 \pm 0.02$ | $0.54 \pm 0.08$ | $0.65 \pm 0.04$ | $0.58 \pm 0.05$ | $0.49 \pm 0.04$ | $0.61 \pm 0.03$ |
| PaLM2-M | $0.58 \pm 0.02$ | $0.62 \pm 0.06$ | $0.59 \pm 0.04$ | $0.60 \pm 0.05$ | $0.70 \pm 0.03$ | $0.56 \pm 0.04$ | $0.46 \pm 0.04$ | $0.62 \pm 0.05$ |
| PaLM2-L | $0.53 \pm 0.05$ | $0.60 \pm 0.05$ | $0.61 \pm 0.05$ | $0.56 \pm 0.03$ | $0.62 \pm 0.02$ | $0.60 \pm 0.07$ | $0.42 \pm 0.03$ | $0.70 \pm 0.03$ |

Table 5: Self-similarity exponent S evaluated on the first 2048 tokens, after trimming the first 100 tokens, of documents belonging to each of the shown domains. Only documents with a minimum length of 4K tokens are used.

| Model | OpenWebText2 | Github | FreeLaw | Pile-CC | Wikipedia | PubMed | Mathematics | ArXiv |
|---|---|---|---|---|---|---|---|---|
| T5-Decoder-110M | 0.63 ± 0.00 | 0.82 ± 0.01 | 0.62 ± 0.01 | 0.67 ± 0.01 | 0.62 ± 0.01 | 0.65 ± 0.00 | 0.54 ± 0.01 | 0.68 ± 0.01 |
| T5-Decoder-340M | 0.63 ± 0.01 | 0.82 ± 0.01 | 0.62 ± 0.00 | 0.67 ± 0.00 | 0.62 ± 0.01 | 0.64 ± 0.01 | 0.54 ± 0.00 | 0.67 ± 0.01 |
| T5-Decoder-1B | 0.63 ± 0.01 | 0.83 ± 0.01 | 0.63 ± 0.01 | 0.67 ± 0.00 | 0.62 ± 0.01 | 0.64 ± 0.00 | 0.54 ± 0.00 | 0.67 ± 0.00 |
| T5-Decoder-5B | 0.63 ± 0.01 | 0.82 ± 0.00 | 0.62 ± 0.01 | 0.67 ± 0.01 | 0.62 ± 0.01 | 0.64 ± 0.01 | 0.54 ± 0.00 | 0.67 ± 0.00 |
| PaLM1-8B | 0.65 ± 0.01 | 0.81 ± 0.01 | 0.66 ± 0.00 | 0.68 ± 0.01 | 0.66 ± 0.00 | 0.65 ± 0.01 | 0.57 ± 0.00 | 0.69 ± 0.01 |
| PaLM1-62B | 0.66 ± 0.01 | 0.80 ± 0.00 | 0.67 ± 0.01 | 0.69 ± 0.01 | 0.68 ± 0.00 | 0.65 ± 0.00 | 0.57 ± 0.00 | 0.70 ± 0.00 |
| PaLM1-540B | 0.67 ± 0.00 | 0.79 ± 0.01 | 0.68 ± 0.00 | 0.69 ± 0.01 | 0.71 ± 0.01 | 0.65 ± 0.01 | 0.56 ± 0.00 | 0.70 ± 0.01 |
| PaLM2-XXS | 0.65 ± 0.01 | 0.81 ± 0.01 | 0.65 ± 0.01 | 0.68 ± 0.01 | 0.66 ± 0.01 | 0.65 ± 0.01 | 0.58 ± 0.00 | 0.71 ± 0.01 |
| PaLM2-XS | 0.65 ± 0.01 | 0.81 ± 0.01 | 0.66 ± 0.01 | 0.68 ± 0.01 | 0.67 ± 0.00 | 0.65 ± 0.00 | 0.56 ± 0.01 | 0.71 ± 0.01 |
| PaLM2-S | 0.67 ± 0.01 | 0.80 ± 0.01 | 0.66 ± 0.01 | 0.69 ± 0.00 | 0.68 ± 0.01 | 0.65 ± 0.01 | 0.54 ± 0.00 | 0.71 ± 0.00 |
| PaLM2-M | 0.67 ± 0.01 | 0.80 ± 0.01 | 0.67 ± 0.01 | 0.70 ± 0.01 | 0.70 ± 0.01 | 0.65 ± 0.01 | 0.52 ± 0.01 | 0.72 ± 0.01 |
| PaLM2-L | 0.68 ± 0.01 | 0.79 ± 0.01 | 0.68 ± 0.00 | 0.70 ± 0.00 | 0.74 ± 0.01 | 0.65 ± 0.00 | 0.50 ± 0.01 | 0.72 ± 0.01 |

Table 6: Hurst exponent H evaluated on the first 2048 tokens, after trimming the first 100 tokens, of documents belonging to each of the shown domains. Only documents with a minimum length of 4K tokens are used.

| Model | OpenWebText2 | Github | FreeLaw | Pile-CC | Wikipedia | PubMed | Mathematics | ArXiv |
|---|---|---|---|---|---|---|---|---|
| T5-Decoder-110M | 0.44 ± 0.01 | 0.53 ± 0.00 | 0.42 ± 0.00 | 0.49 ± 0.01 | 0.45 ± 0.00 | 0.43 ± 0.00 | 0.33 ± 0.00 | 0.45 ± 0.00 |
| T5-Decoder-340M | 0.44 ± 0.02 | 0.53 ± 0.00 | 0.43 ± 0.00 | 0.49 ± 0.00 | 0.45 ± 0.01 | 0.43 ± 0.00 | 0.33 ± 0.00 | 0.45 ± 0.00 |
| T5-Decoder-1B | 0.43 ± 0.01 | 0.53 ± 0.00 | 0.43 ± 0.01 | 0.49 ± 0.01 | 0.45 ± 0.01 | 0.42 ± 0.00 | 0.33 ± 0.00 | 0.45 ± 0.01 |
| T5-Decoder-5B | 0.43 ± 0.01 | 0.53 ± 0.00 | 0.44 ± 0.00 | 0.49 ± 0.01 | 0.45 ± 0.00 | 0.42 ± 0.00 | 0.34 ± 0.00 | 0.45 ± 0.00 |
| PaLM1-8B | 0.45 ± 0.00 | 0.51 ± 0.00 | 0.46 ± 0.00 | 0.49 ± 0.01 | 0.48 ± 0.01 | 0.44 ± 0.01 | 0.34 ± 0.00 | 0.48 ± 0.01 |
| PaLM1-62B | 0.45 ± 0.00 | 0.50 ± 0.01 | 0.47 ± 0.00 | 0.49 ± 0.01 | 0.49 ± 0.00 | 0.44 ± 0.00 | 0.33 ± 0.00 | 0.48 ± 0.01 |
| PaLM1-540B | 0.46 ± 0.01 | 0.49 ± 0.01 | 0.47 ± 0.00 | 0.50 ± 0.01 | 0.50 ± 0.00 | 0.44 ± 0.00 | 0.33 ± 0.01 | 0.48 ± 0.00 |
| PaLM2-XXS | 0.44 ± 0.01 | 0.50 ± 0.00 | 0.45 ± 0.00 | 0.50 ± 0.01 | 0.48 ± 0.00 | 0.45 ± 0.00 | 0.34 ± 0.00 | 0.49 ± 0.00 |
| PaLM2-XS | 0.45 ± 0.01 | 0.50 ± 0.01 | 0.46 ± 0.01 | 0.49 ± 0.00 | 0.48 ± 0.00 | 0.44 ± 0.00 | 0.33 ± 0.01 | 0.49 ± 0.00 |
| PaLM2-S | 0.45 ± 0.00 | 0.49 ± 0.00 | 0.47 ± 0.00 | 0.50 ± 0.01 | 0.50 ± 0.01 | 0.44 ± 0.00 | 0.31 ± 0.00 | 0.49 ± 0.00 |
| PaLM2-M | 0.45 ± 0.01 | 0.49 ± 0.01 | 0.48 ± 0.01 | 0.50 ± 0.01 | 0.50 ± 0.00 | 0.44 ± 0.00 | 0.29 ± 0.00 | 0.49 ± 0.01 |
| PaLM2-L | 0.46 ± 0.01 | 0.49 ± 0.00 | 0.49 ± 0.00 | 0.50 ± 0.00 | 0.52 ± 0.00 | 0.44 ± 0.00 | 0.28 ± 0.00 | 0.49 ± 0.00 |

Table 7: Joseph exponent J evaluated on the first 2048 tokens, after trimming the first 100 tokens, of documents belonging to each of the shown domains. Only documents with a minimum length of 4K tokens are used.

## C Predicting Downstream Performance

Table 8 presents detailed downstream performance results, along with corresponding upstream metrics.

In Table 9, we repeat the same analysis in Section 3 using the adjusted $R^2$ coefficient, but with the self-similarity S and Joseph exponents J. Unlike in the median Hurst exponent, we do not observe

any improvement when combining perplexity scores with the self-similarity exponent S or the Joseph exponent J.

| Model | BPB | 0S BBH Direct | 0S BBH CoT | 0S MMLU | 3S BBH Direct | 3S BBH CoT | 5S MMLU | 8S GSM8K CoT | 0S BBH +MMLU | FS BBH +MMLU +GSM8K |
|---|---|---|---|---|---|---|---|---|---|---|
| T5-Decoder-110M | 1.11 | 0.83 | 0.11 | 25.65 | 21.36 | 5.69 | 25.62 | 0.91 | 13.06 | 13.35 |
| T5-Decoder-340M | 1.00 | 0.96 | 0.17 | 25.72 | 23.57 | 10.03 | 25.98 | 1.59 | 13.14 | 14.79 |
| T5-Decoder-1B | 0.92 | 1.29 | 0.14 | 25.99 | 24.26 | 13.19 | 24.82 | 1.14 | 13.35 | 14.90 |
| T5-Decoder-5B | 0.85 | 2.13 | 0.48 | 24.41 | 24.76 | 18.05 | 25.63 | 2.20 | 12.86 | 16.41 |
| PaLM1-8B | 0.78 | 6.46 | 1.21 | 23.53 | 32.18 | 27.60 | 24.56 | 5.16 | 13.68 | 19.87 |
| PaLM1-62B | 0.70 | 13.79 | 0.83 | 51.86 | 39.51 | 39.70 | 54.78 | 29.57 | 29.59 | 41.32 |
| PaLM1-540B | 0.66 | 23.26 | 4.72 | 67.78 | 52.44 | 56.02 | 70.50 | 56.79 | 40.89 | 60.51 |
| PaLM2-XXS | 0.81 | 8.99 | 0.13 | 25.26 | 30.71 | 26.08 | 24.72 | 2.96 | 14.91 | 18.69 |
| PaLM2-XS | 0.73 | 16.68 | 0.95 | 49.69 | 38.28 | 37.64 | 47.42 | 22.14 | 29.25 | 35.84 |
| PaLM2-S | 0.67 | 23.60 | 4.24 | 69.89 | 48.88 | 50.88 | 68.12 | 50.49 | 41.91 | 56.16 |
| PaLM2-M | 0.65 | 21.32 | 5.70 | 69.62 | 52.49 | 56.04 | 69.33 | 59.21 | 41.57 | 60.94 |
| PaLM2-L | 0.61 | 24.00 | 10.19 | 79.10 | 66.34 | 66.66 | 78.64 | 80.36 | 48.10 | 75.17 |

Table 8: Full downstream few-shot evaluation results compared to upstream BPB. Here, BPB is computed over The Pile validation split using the first 2048 tokens of every document. All evaluation results are reported as raw (un-normalized) accuracy.

Please note that our results are not directly comparable to all previous published results for the same models; please cite the original results from [12, 6]. Here, we only aim for a fair comparison between models: only pretrained models without instruction tuning are used, we do not optimize any prompts for each model, and we evaluate all models using only a 2K sequence length.

| | BPB | S | J | BPB+S | BPB+J |
|---|---|---|---|---|---|
| 0S BBH Direct | 0.785 | -0.060 | 0.673 | 0.761 | 0.794 |
| 0S MMLU | 0.653 | -0.067 | 0.426 | 0.614 | 0.614 |
| 0S BBH+MMLU | 0.685 | -0.065 | 0.472 | 0.650 | 0.651 |
| 3S BBH Direct | 0.767 | -0.030 | 0.599 | 0.744 | 0.754 |
| 3S BBH CoT | 0.881 | -0.026 | 0.678 | 0.870 | 0.879 |
| 5S MMLU | 0.660 | -0.044 | 0.421 | 0.624 | 0.622 |
| 8S GSM8K CoT | 0.654 | -0.037 | 0.427 | 0.619 | 0.616 |
| FS BBH + MMLU+GSM8K | 0.717 | -0.036 | 0.489 | 0.687 | 0.686 |

Table 9: Adjusted $R^2$, which measures the proportion of variation in downstream performance (row) that is predictable from the given input(s) (column) using a trained linear regressor. Unlike in the median Hurst exponent, we do not observe any improvement when combining BPB scores with the self-similarity exponent S or the Joseph exponent J.

## D   Gemma-2B Results

In this section, we present empirical results using the publicly released Gemma-2B checkpoint [65]. Results are shown in Figures 8 and 9.

Gemma-2B is much smaller than all of the models we have used previously. Yet, we generally observe similar conclusions. First, we have a self-similar structure with near perfect linear fits in a log-log plots. Second, we also observe power laws using the rescaled-range analysis, with a Hurst exponent of about 0.7 in most domains except DM Mathematics (smallest Hurst exponent of about 0.58) and GitHub (largest Hurst exponent of about 0.82), in general agreement to the rest of the models. Both the self-similarity and Hurst parameters are provided in Table 10.

## E   Comparison to n-Gram Models

Here, we report some early investigations that examine how fractal parameters change as one includes longer contexts during inference. The setup used in this experiment is slightly different from what we use throughout the rest of the paper. Specifically, we take the Wikipedia dataset

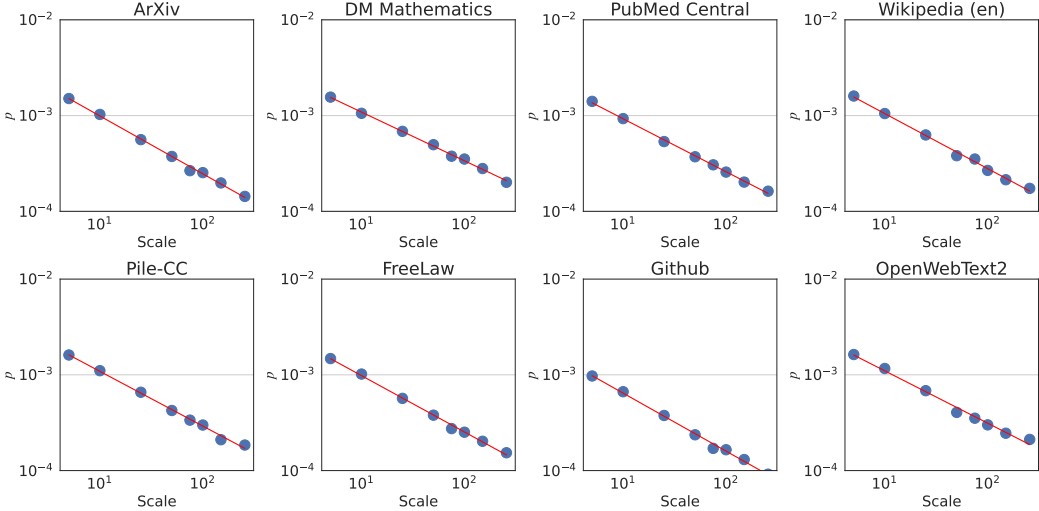

Figure 8: Here, we follow a similar setup to Figure 2, but using the publicly released Gemma-2B checkpoint [65], where the $y$-axis is the peak probability. We continue to observe linear fits in a log-log scale over at least two orders of magnitude, thus confirming a self-similar structure.

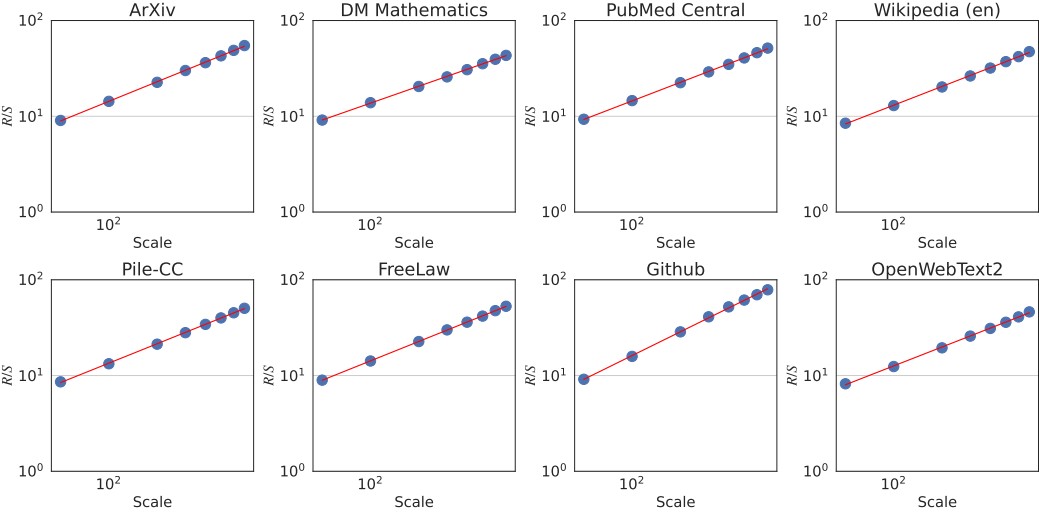

Figure 9: Here, we follow a similar setup to Figure 3, but using the publicly released Gemma-2B checkpoint [65], where the $y$-axis is the rescaled-range (R/S). We continue to observe linear fits in a log-log scale over at least two orders of magnitude.

(`wikipedia/20230601.en`) dataset [67]. We split each document of length >4K words along *word boundaries* and constrain the context length during inference in PaLM-8B to $n$ words, for $n \in \{1, 16, 32, 64, 128, 256, 512, 1024, 2048\}$. Hence, the language model now resembles an $n$-gram model. We calculate probability scores and calculate fractal parameters accordingly. Figure 10 shows how both the self-similarity exponent S and the Hurst parameter H change as a function of $n$.

We observe that H increases monotonically with context length as expected, since it implies more predictability. However, even in a 1-gram model, H can be larger than 1/2 because the words themselves are not independent of each other, so the sequence of probability scores can still contain dependence over time in a 1-gram model.

| OpenWebText2 | Github | FreeLaw | Pile-CC | Wikipedia | PubMed | Mathematics | ArXiv |
|---|---|---|---|---|---|---|---|
| S | | | | | | | |
| $0.55 \pm 0.02$ | $0.63 \pm 0.01$ | $0.60 \pm 0.02$ | $0.58 \pm 0.02$ | $0.57 \pm 0.02$ | $0.57 \pm 0.02$ | $0.50 \pm 0.02$ | $0.61 \pm 0.02$ |
| H | | | | | | | |
| $0.65 \pm 0.01$ | $0.83 \pm 0.01$ | $0.67 \pm 0.01$ | $0.68 \pm 0.01$ | $0.65 \pm 0.01$ | $0.64 \pm 0.01$ | $0.58 \pm 0.01$ | $0.69 \pm 0.01$ |

Table 10: Self-similarity exponent (S) and the Hurst parameter (H) when using Gemma-2B. These values compare well to the ones obtained using the other models.

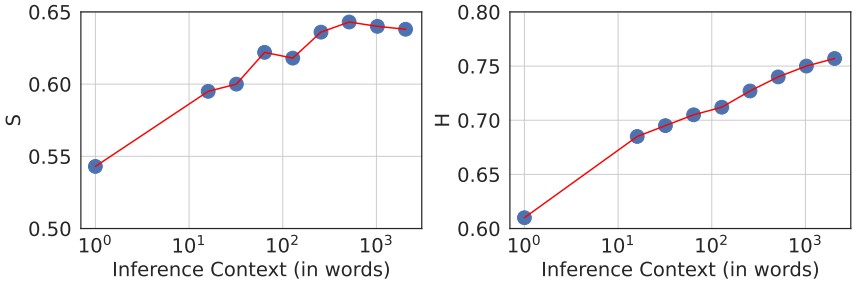

Figure 10: S and H plotted for different constructions of bits, as we vary the prefix length during inference.

