# OpenReview forum: "Fractal Patterns May Illuminate the Success of Next-Token Prediction"
_NeurIPS.cc/2024/Conference — NeurIPS 2024 poster_

### Official Review · Reviewer_ND2C · 2024-07-12

**Soundness:** 3
**Presentation:** 3
**Contribution:** 4
**Rating:** 7
**Confidence:** 3

**Summary:**

This paper provides a detailed application of ideas from fractal geometry to natural language data, using language models to compute the relevant information-theoretic properties.  They find, in particular, tell-tale evidence of self-similarity (common structure across scales) and long-range dependencies.  A particularly interesting result, in my opinion, is that using some of the estimated fractal parameters can help predict LMs' downstream performance over and above what can be predicted from raw LM performance (bits-per-byte), suggesting that some of this performance can be explained by their discovery of this fractal structure.  While the paper can be a bit dense and contains a huge range both of theoretical background and experimental results that make it a little hard to read, I do believe that it pays off to understand and makes a nice contribution that will be of interest to many people in the field.

**Strengths:**

* Provides a thorough and detailed application of fractal geometry to natural language data.
* Use of the new fractal metric helps predict downstream task performance more than just BPB alone.
* Shows that the main findings are robust to the choice of LLM that is used for estimating information-theoretic properties of text.

**Weaknesses:**

* Could cite more literature about e.g. dependency lengths (https://doi.org/10.1073/pnas.1502134112) and other known properties of natural language.  For instance, "duality of patterning" (https://doi.org/10.1515/langcog-2012-0015) has been used to refer to the fact that morphemes->words and words->sentences exhibit similar structural properties, which is akin to self-similarity.
* Although the paper is mostly self-contained, it can be very dense for readers not very familiar with the mathematics of fractals.

**Questions:**

* Lines 46-47: if the reader is not familiar with S, it's not clear what the value means and why it has the implications that it does.  Can you explain more at this point or move these remarks to a bit later?
* Have you checked whether a power-law is genuinely the best fit to the data, versus other laws that often resemble them (e.g. linear in log-log)? See e.g. https://doi.org/10.1126/science.1216142
* Can you say more about the use of LLMs to compute and represent the information-theoretic quantities in this context?  As a way of making sure that we are learning about language in particular and not just artifacts/properties of the models.

**Limitations:**

Yes; I appreciated especially the mention of the results being English-only

---

> ### Author Rebuttal · Authors · 2024-08-06
>
> We would like to thank the reviewer for the careful and insightful feedback. We especially appreciate the positive assessment of the thoroughness, originality, and importance of our work. We hope that our rebuttal below addresses all of the reviewer’s questions, are happy to provide more details, and look forward to the reviewer’s response to our rebuttal.
>
> **Additional References**
>
> Thank you for the great suggestions. We will add those references along with a brief discussion in the related works section.
>
> **Clarity of writing**
>
> We will add more clarity in the revised version of the paper. If there are any specific suggestions for restructuring or clarifying some points, please let us know and we would gladly incorporate them into the paper.
> We will add more explanation about S and what it means prior to the discussion in Lines 46-47. Informally, a smaller value of S indicates a slower decay in the auto-correlation function and more fractal structure.
>
> **Power law fit**
>
> This is an important point and we appreciate the suggested reference. In our experiments, we observe a near perfect linear fit on a log-log plot over more than two orders of magnitude (Figures 2 and 3), in agreement with what (Stumpf and Porter, 2012) argued for. (Stumpf and Porter, 2012) wrote: “As a rule of thumb, a candidate power law should exhibit an approximately linear relationship on a log-log plot over at least two orders of magnitude in both the x and y axes.” This is exactly what we observe. We will clarify this point in the revised version of the paper.
>
> **Information-theoretic complexity**
>
> Thank you for raising this point. Please see our general comment above on why we believe an information-theoretic complexity is meaningful for our analysis. We will clarify this more in the revised version of the paper.
>
> **Follow-up**
>
> We are grateful again for your detailed and constructive feedback. If we have satisfactorily answered your questions, we hope you would consider revising your scores. Otherwise, please let us know if there are any other questions or concerns, so we can respond to them during the discussion period.

---

> > ### Author Response · Authors · 2024-08-11
> > **Follow up**
> >
> > Dear reviewer,
> >
> > Thank you again for the detailed and constructive comments.
> >
> > As the discussion period is about to end, we would like to ensure we've addressed all of your questions and concerns. If you feel we have satisfactorily responded, please let us know. If you have any further questions, we are happy to address them before the discussion period closes.
> >
> > Sincerely

---

> > > ### Comment · Reviewer_ND2C · 2024-08-12
> > >
> > > Thank you!  And apologies for my delayed response.  I am happy with the response and your engagement with the other reviews as well.  I would like to hear more about "why we believe an information-theoretic complexity is meaningful", and the use of LLMs for that measurement, but it's by no means a blocker for me, and I would like to see the paper accepted.

---

### Official Review · Reviewer_cohu · 2024-07-12

**Soundness:** 3
**Presentation:** 2
**Contribution:** 3
**Rating:** 6
**Confidence:** 2

**Summary:**

This paper introduce a new perspective to that language is self-similar and predictability and self-similarity together imply long-range dependency, based on empirical analysis across different scales of LMs and information theoretic views. This new perspective may enable us to understand the strong capabilities of current causal LLMs. They studies three parameters, self-similarity (Hölder) exponent, the Hurst parameter, and the fractal dimension, as well as Joseph exponent. They also introduces a new metric that can more precisely approximately downstream performance than BPB.

**Strengths:**

- This paper provides some new interesting perspectives to advance our understanding of how LLMs acquire such strong capabilities after large-scale pre-training using simple autoregressive training objective.
- They conduct large-scale analysis using LMs with different scale in three different model families.
- They also empirically found that a median Hurst exponent can be a more reliable indicator of downstream performance than perplexity-based BPB. While perplexity has shown to strongly correlated with downstream performance, prior studies also show their limitations on predicting downstream performance, and this work may encourage future work to use this new metric instead.

**Weaknesses:**

- To my understanding, this work views language  as a  mere sequence of negative log probabilities while prior studies in linguistic / NLP often consider much more rich structures of languages and am not fully convinced the validity of analysis.
- While the analysis includes three models, none of the models’ checkpoints aren’t publicly available (i.e., PaLM and PaLM2 and their newly trained T5-decoder only models) and followup work may not be able to reproduce the results. I’m curious why the authors didn’t test other models with openly available model checkpoints such as Llama 2, 3 / OLMo / Pythia.
- Overall I found the paper is a bit hard to follow (e.g., Introduction discusses prior literature in depth before providing high-level overview or motivation, Experimental setups come write after preliminaries). Some restructures of sections or writing may improves readability of the paper.

**Questions:**

- Did authors try other models with publicly available model checkpoints so that people can reproduce the results?
- Why did authors train T5-decoder model from scratch, instead of using T5 variant that went through next token prediction training (e.g., T5 LM adapt)?

**Limitations:**

The authors provide limitation sections that include some of the limitations I was thinking about.

---

> ### Author Rebuttal · Authors · 2024-08-06
>
> We would like to thank the reviewer for the careful and insightful feedback. We especially appreciate the positive assessment of the thoroughness, originality, and importance of our work. We hope that our rebuttal below addresses all of the reviewer’s questions, are happy to provide more details, and look forward to the reviewer’s response to our rebuttal.
>
> **Information-theoretic complexity**
>
> Thank you for raising this point. Please see our general comment above on why we believe an information-theoretic complexity is meaningful for our analysis. We will clarify this more in the revised version of the paper.
>
>
>
> **Reproducibility**
>
> This is an important point and we thank the reviewer for raising it. Our response is threefold.
>
> First, we have run new experiments using the released gemma-2b checkpoint (https://huggingface.co/google/gemma-2b) to compare it with the models used in the paper. Due to the short rebuttal time, we were able to use the smallest gemma model and a subset of the Pile validation split only. But, as discussed in the general comment above, we observe a good agreement overall. Gemma 2B is much smaller than all of the models we have used in our paper. Yet, we generally observe similar conclusions. Please see our general response above for details. Based on these findings, we do believe our results will continue to hold using publicly available checkpoints. We plan to continue running the analysis on the full validation data split of the Pile and include those results in the supplementary materials of the paper.
>
> Second, the reason we included T5 trained from scratch is to ensure that our findings are reproducible. We believe that by training a model from scratch instead of using released checkpoints, we are more confident in our findings. We have provided all of our training details for reproducibility. We hope this answers your question.
>
> Third, to ensure reproducibility and to encourage the community to explore these fractal properties in language, we also release a code for calculating all fractal params.
>
> **Clarity of writing**
>
> We will add more clarity in the revised version of the paper. If there are any specific suggestions for restructuring or clarifying some points, please let us know and we would gladly incorporate them into the paper.
>
> **Follow-up**
>
> We are grateful again for your detailed and constructive feedback. If we have satisfactorily answered your questions, we hope you would consider revising your scores. Otherwise, please let us know if there are any other questions or concerns, so we can respond to them during the discussion period.

---

> > ### Author Response · Authors · 2024-08-11
> > **Follow up**
> >
> > Dear reviewer,
> >
> > Thank you again for the detailed and constructive comments.
> >
> > As the discussion period is about to end, we would like to ensure we've addressed all of your questions and concerns. If you feel we have satisfactorily responded, please let us know. If you have any further questions, we are happy to address them before the discussion period closes.
> >
> > Sincerely

---

> > > ### Comment · Reviewer_cohu · 2024-08-11
> > > **Thank you for your response. I'll keep my score.**
> > >
> > > Thank you for your response. The new results based on gemma-2b look good. I'll keep my score. My original concern about reproducibility is mostly based on the initial evaluations are based on either closed models or the authors' checkpoints that are not publicly released. Adding gemma 2B mostly addressed this concern.

---

### Official Review · Reviewer_iLZg · 2024-07-13

**Soundness:** 2
**Presentation:** 3
**Contribution:** 2
**Rating:** 5
**Confidence:** 2

**Summary:**

The paper draws connections between fractal patterns and language by evaluating properties such as self-similarity and long-range-dependency. Using a range of LLMs, they estimate the Holder exponent, Hurst parameter and fractal dimension for language from different domains, including web text, code, and math problems. They show that these parameters may have connections with LLM learning ability, demonstrating that using the median Hurst parameter can improve prediction of downstream model performance over only using bits-per-byte.

**Strengths:**

- The authors estimate fractal parameters across a range of datasets and LLMs, showing that they are fairly robust to choice of LLM and domain. For domains with significant deviation, such as DM-Mathematics, the authors are able to attribute this to the dataset's lack of long-range dependency.
- The authors show that the Hurst parameter captures useful information for predicting downstream performance that is not captured by BPB alone.

**Weaknesses:**

- Even though fractal parameters were found to be consistent across LLMs, I think their stated contribution (this 'establish[es] that language is self-similar and long-range-dependent') is far too strong a claim. As the authors mention, their method ignores many important facets of language, such as semantic nuance, and is reliant on the current state of LLMs' ability to model language.
- I am unclear how future work can build upon these insights about language, especially that 'exploiting self-similarity more directly' could lead to further LLM optimization.

**Questions:**

I don't follow the potential connection between self-similarity and the success of parameter sharing? (mentioned in Limitations section)

Typos:
- Line 131: such bits-per-byte (BPB)
- Line 132: BPB is a widely used as a
- Line 272: such as One example

**Limitations:**

Yes, the authors discuss that their analysis is limited to English data and fails to capture the semantic meaning of language. However, as mentioned in Questions section, the possible connection between self-similarity and parameter-sharing is not obvious to me and could be further explained.

---

> ### Author Rebuttal · Authors · 2024-08-06
>
> We would like to thank the reviewer for the careful and insightful feedback. We especially appreciate the positive assessment of the thoroughness, originality, and importance of our work. We hope that our rebuttal below addresses all of the reviewer’s questions, are happy to provide more details, and look forward to the reviewer’s response to our rebuttal.
>
> **Information-theoretic complexity**
>
> Thank you for raising this point. Please see our general comment above on why we believe an information-theoretic complexity is meaningful for our analysis. We will clarify this more in the revised version of the paper.
>
> **Future work and parameter sharing**
>
> We are currently exploring several directions for future work. For example, do texts generated by LLMs have a self-similar structure? What is the impact of RLHF on the self-similar structure of generated texts? Can we develop techniques during training to encourage models to mimic the self-similarity we see in language?
>
> On the architecture side, self-similarity in language means that linguistic patterns repeat at different scales. This mirrors how parameter sharing works: the same set of parameters (attention mechanisms) is applied across different positions in a sequence, assuming similar operations are needed to understand the language at different levels. This is what we mean by saying that the success of parameter sharing techniques, such as ALBERT, may be partially explained by self-similarity. We will explain this more in the revised version of the paper. We have some preliminary experiments, for instance, that show that parameter-sharing is more effective for language than in vision (presumably because language is self-similar). But, it is unclear yet what the best way to exploit self-similarity in the architecture is. We believe these are exciting areas for future research.
>
> **Typos**
>
> Thank you for pointing them out. We will fix them in the revised version of the paper.
>
> **Follow-up**
>
> We are grateful again for your detailed and constructive feedback. If we have satisfactorily answered your questions, we hope you would consider revising your scores. Otherwise, please let us know if there are any other questions or concerns, so we can respond to them during the discussion period.

---

> > ### Author Response · Authors · 2024-08-11
> > **Follow up**
> >
> > Dear reviewer,
> >
> > Thank you again for the detailed and constructive comments.
> >
> > As the discussion period is about to end, we would like to ensure we've addressed all of your questions and concerns. If you feel we have satisfactorily responded, please let us know. Otherwise, please let us know your remaining concerns so we can address them before the discussion period closes.
> >
> > Sincerely

---

> > ### Comment · Reviewer_iLZg · 2024-08-12
> >
> > Thanks for your response and clarifications. I will maintain my current score.

---

> > > ### Author Response · Authors · 2024-08-12
> > > **Thank you**
> > >
> > > Thank you for engaging with us. We would appreciate it if you let us know of any remaining concerns so we can respond to them.
> > >
> > > With regard to the "poor" contribution rating, we would like to respectfully emphasize the novelty and technical quality of our work.
> > >
> > > - We believe our work is *quite original*: we are not aware of any prior work that has attempted to study the fractal nature of language using LLMs. We offer a quantitative evidence for the self-similar and long-range dependent nature of language.
> > >
> > > - We demonstrate an *interesting* connection between the Hurst parameter and downstream performance of LLMs. This is by no means obvious or could have been expected in advance.
> > >
> > > - We have conducted an *extensive* empirical analysis across various domains and architectures, showcasing the robustness of our findings. This includes downstream evaluations on popular benchmarks that cover many tasks (please see our response above), prompted using various strategies (e.g. direct and chain-of-thought with either 0, 3, 5, or 8 shots).
> > >
> > > - Our work is *self-contained*: we provide the necessary mathematical background to explain how to calculate all fractal parameters, and we have strived to make it easy to read and follow.
> > >
> > > - We include all details necessary to reproduce the results in the supplementary materials, in addition to releasing the code for calculating fractal parameters. We are also including results using gemma-2b as requested by reviewer cohu.
> > >
> > > We agree that our work has some limitations, as one cannot answer all questions in a single paper (e.g. we focus on the English language alone). But, we respectfully disagree that the contribution of this work be considered "poor". We would appreciate it if you let us know of any remaining concerns so we can respond to them during the remainder of the discussion period.
> > >
> > > Sincerely

---

> > > > ### Comment · Reviewer_iLZg · 2024-08-12
> > > >
> > > > Thanks for emphasizing your contributions: after reading through the material provided during rebuttal (particularly your additional n-gram experiments for reviewer UXZb), I agree with the points you make and I will raise my score to 5 and change contribution to 2 (fair). However, I'd appreciate if you could clarify one thing about the Palm-8B n-gram experiment: for your smallest prefix-length setting (16-gram?), you observe a Hurst exponent of ~68.5. I do not have a strong intuition for sensible values, but it seems that this is still quite high? Especially because as you mention, we would not expect n-gram models, where n is small, to generate outputs with LRD. Would it be correct to assume that for a very small n-gram model (unigram/bigram/4-gram), we would expect Hurst parameter close to 0.5, and if so, is this what we observe?
> > > >
> > > > Aside from this, I also have a suggestion for clarity: for the revised version of the paper, I would appreciate if you could provide alternative versions of Figure 7 for the Appendix. I'd like separate plots where downstream performance (currently represented by bubble size) is plotted against median BFP/Hurst/Hb. This would make it easier to qualitatively evaluate the relationship between these values.

---

> > > > > ### Author Response · Authors · 2024-08-13
> > > > > **Thank you**
> > > > >
> > > > > Dear reviewer,
> > > > >
> > > > > Thank you very much for the suggestions and the continued engagement.
> > > > >
> > > > > - We indeed expect that the Hurst exponent would be close to 0.5 for unigram/bigram models. We will add those to the figure as well.
> > > > >
> > > > > - As requested, we will add separate plots in the appendix for downstream performance plotted against the median Hurst parameter.
> > > > >
> > > > > Sincerely

---

### Official Review · Reviewer_UXZb · 2024-07-13

**Soundness:** 2
**Presentation:** 2
**Contribution:** 2
**Rating:** 7
**Confidence:** 3

**Summary:**

In this paper, the authors try to reveal the existence of fractal structures to handle language in language modeling using recent language models based on the next token prediction. For that purpose, the authors rely on several aspects of fractal structures, which are self-similarity, long-range dependence, and information-theoretic complexity. The authors describe these aspects using metrics for typical phenomena under the assumption of fractal structure, which are self-similarity exponent, Hurst parameter, fragmental dimension, and Joseph effect. The experimental results show that the characteristics of modeling texts of ArXiv, Github, and Wikipedia data in The Pile validation split, whose tokens are longer than 4K, are along with their assumption. In the analysis of the downstream tasks, BBH, MMLU, and GSM8K also show that the task-solving performances are predictable from the language modeling performance based on their assumption, except for considering sequence lengths in training.

**Strengths:**

- Providing an assumption to unify characteristics of interpreting language and language modeling under the variously used approach, next token prediction may help solving various kinds of tasks related to natural languages.
- The analysis is not only restricted to language modeling and the authors investigate the performance correlation between language modeling and downstream tasks based on their assumption.

**Weaknesses:**

- The target language is limited to English. Thus, the validity of expanding the insights in the paper to other languages is uncertain.
- The assumed baseline in this analysis is a random sequence. The authors can use finite-state automatons (FSAs) or Context-free grammars (CFGs) to make sequences that are random but close to languages. Moreover, to focus on the success of recent pre-trained language models, you should have prepared N-gram language models as baselines.
- How strongly the observation results fit the assumed distribution is not calculated mathematically, like model selection by Information Criteria such as Akaike's Information Criteria (AIC) and Bayesian information criteria (BIC).
- There is a gap between the ability to mimic the characteristics of language and showing the required knowledge for questions. Thus, discussing the correlation between the performance of language modeling and downstream tasks is limited from this viewpoint. Hence, targeting generation tasks like summarization, story generation, and machine translation are more suitable for deepening the analysis.

**Questions:**

- In the analysis, the authors seem to only focus on the context length in training even though the inference of long sequential tokens is closer to the aspect of long distant dependencies between tokens in language. What is the main reason for this decision?

[Comment] In presentation style, instead of showing the method name, showing the motivation or target phenomenon as the section or paragraph title, like "Joseph effect" -> "Burstiness," may support the reader's understanding.

**Limitations:**

The limitation does not include the gap between language modeling and solving downstream tasks.

---

> ### Author Rebuttal · Authors · 2024-08-06
>
> We would like to thank the reviewer for the careful and insightful feedback. We hope that our rebuttal below addresses all of the reviewer’s questions, are happy to provide more details, and look forward to the reviewer’s response to our rebuttal.
>
> **The baseline**
>
> The baseline we use in Figure 1 is used for illustration purposes only, mainly to show how a self-similar process looks like. We do not use random sequences in our analysis. While we appreciate the suggestion of using n-gram models, our work focuses on the self-similar structure of natural language. Note in particular that the output of an n-gram model, by definition, does *not* have long-range dependence (LRD). So, conducting the same analysis on the output of n-gram models would not yield meaningful insights into the fractal structure of language, as n-gram models cannot generate long-range dependencies.
>
> **Empirical fit**
>
> Thank you for raising this point. We do demonstrate how well the power law fits the actual distribution in Figures 2 and 3. We observe a perfect linear fit in a log-log plot over at least two orders of magnitude. In addition, since the power law is of the form $\beta n^{-c}$, there are only two parameters to estimate ($\beta$ and $c$) so AIC is also quite low, given the near perfect fit and the small number of parameters.
>
> **Evaluation**
>
> We use three popular and quite diverse benchmarks (BBH, MMLU, GSM8K) and we also try various prompting strategies (e.g. direct and chain-of-thought with 0, 3, 5, 8 shots). These benchmarks are quite diverse and include tasks such as logical deduction, multi-step arithmetic, disambiguation, as well as general knowledge (e.g. history, computer science, law, sports, movies, and dates, etc). In line with your suggestions, they also include translation-related tasks, such as Translation Error Detection in BBH. So, they cover a broad spectrum of tasks. We hope this addresses your concern about the evaluation datasets, and we will clarify this in the revised version of the paper.
>
> **Context length**
>
> The main reason for including the discussion about the context length at training time in our analysis is because our findings suggest one intriguing possibility: that language models might benefit from being trained on long contexts even if short contexts are used at inference time. However, we did not find any evidence yet to support this. We chose to include this in the analysis section because we believe that mentioning negative results would still be very valuable to our community.
>
> **Presentation**
>
> Thank you for the great suggestion about the presentation style. We will definitely do that in the revised version of the paper.
>
>
> **Follow-up**
>
> We are grateful again for your detailed and constructive feedback. If we have satisfactorily answered your questions, we hope you would consider revising your scores. Otherwise, please let us know if there are any other questions or concerns, so we can respond to them during the discussion period.

---

> > ### Author Response · Authors · 2024-08-11
> > **Follow up**
> >
> > Dear reviewer,
> >
> > Thank you again for the detailed and constructive comments.
> >
> > As the discussion period is about to end, we would like to ensure we've addressed all of your questions and concerns. If you feel we have satisfactorily responded, please let us know. Otherwise, please let us know your remaining concerns so we can address them before the discussion period closes.
> >
> > Sincerely

---

> > > ### Comment · Reviewer_UXZb · 2024-08-12
> > > **Some concerns are resolved**
> > >
> > > I appreciate your detailed response. These are my replies to your rebuttals:
> > >
> > > >**The baseline**
> > >
> > > > The baseline we use in Figure 1 is used for illustration purposes only, mainly to show how a self-similar process looks like. We do not use random sequences in our analysis. While we appreciate the suggestion of using n-gram models, our work focuses on the self-similar structure of natural language. Note in particular that the output of an n-gram model, by definition, does not have long-range dependence (LRD). So, conducting the same analysis on the output of n-gram models would not yield meaningful insights into the fractal structure of language, as n-gram models cannot generate long-range dependencies.
> > >
> > > Yes, your paper does not have an obvious baseline in terms of model comparison. However, to consider how the distribution observed on language models satisfies the characteristics of fractal patterns, you need to compare language models with other simple sequence generation methods like n-gram or probabilistic context-free grammars (PCFGs). Currently, the discussion is only based on the observed absolute values of generated texts by language models. Thus, readers find it difficult to estimate how the observed characteristics fit the proposed theory about fractal patterns.
> > >
> > > >**Empirical fit**
> > >
> > > > Thank you for raising this point. We do demonstrate how well the power law fits the actual distribution in Figures 2 and 3. We observe a perfect linear fit in a log-log plot over at least two orders of magnitude. In addition, since the power law is of the form , there are only two parameters to estimate ( and ) so AIC is also quite low, given the near perfect fit and the small number of parameters.
> > >
> > > Thank you for clearing that point. I also think that the current result fits your assumption well. However, without relative comparison, readers cannot exclude the possibility that fractal is common in sequences that are not restricted to languages. Therefore, similar to my first question. Preparing a simple sequence generation model like n-gram or PCFGs can support the faithfulness of your claim.
> > >
> > > >**Evaluation**
> > >
> > > > We use three popular and quite diverse benchmarks (BBH, MMLU, GSM8K) and we also try various prompting strategies (e.g. direct and chain-of-thought with 0, 3, 5, 8 shots). These benchmarks are quite diverse and include tasks such as logical deduction, multi-step arithmetic, disambiguation, as well as general knowledge (e.g. history, computer science, law, sports, movies, and dates, etc). In line with your suggestions, they also include translation-related tasks, such as Translation Error Detection in BBH. So, they cover a broad spectrum of tasks. We hope this addresses your concern about the evaluation datasets, and we will clarify this in the revised version of the paper.
> > >
> > > I appreciate your detailed explanation of the BBH benchmarks. This explanation helps me understand that BBH includes translation-related tasks, and my concern about that part is now resolved.
> > >
> > > >**Context length**
> > >
> > > > The main reason for including the discussion about the context length at training time in our analysis is because our findings suggest one intriguing possibility: that language models might benefit from being trained on long contexts even if short contexts are used at inference time. However, we did not find any evidence yet to support this. We chose to include this in the analysis section because we believe that mentioning negative results would still be very valuable to our community.
> > >
> > > The assumption itself is natural. Since the part is not so close to your main subject of the paper, I understand the stance.
> > >
> > > >**Presentation**
> > >
> > > > Thank you for the great suggestion about the presentation style. We will definitely do that in the revised version of the paper.
> > >
> > > I appreciate your generous attitude. I hope many readers will easily understand the paper with the update.
> > >
> > > >**Follow-up**
> > >
> > > > We are grateful again for your detailed and constructive feedback. If we have satisfactorily answered your questions, we hope you would consider revising your scores. Otherwise, please let us know if there are any other questions or concerns, so we can respond to them during the discussion period.
> > >
> > > Since the response actually solves some concerns, I'll raise my score to 5. In addition, if you can make a promise to me about the following requirement, I'll raise the score to 7:
> > >
> > > Requirement: The analysis of relative differences between the sequences generated by language models and those by n-gram or PCFG-based models is included in the Appendix. By showing the result, your claim is grounded on the objective evidence. It changes the reliability of your assumption. Since the change is mainly conducted in the Appendix part, I think this is not included in the major revision. Needless to say, you have no need to follow the requirements during the rebuttal period.

---

> > > > ### Author Response · Authors · 2024-08-12
> > > >
> > > > Dear reviewer
> > > >
> > > > Thank you for engaging with us and for clarifying your question about n-gram models. We have done an early experiment as a sanity check (to verify that H indeed captures LRD) but did not include it in the paper because it was only done on a single model (PaLM-8B) on a single domain (Wikipedia). You can find the results here: https://ibb.co/MhDh9CW and https://ibb.co/Btq94Kz.
> > > >
> > > > In this experiment, we limit the context length of PaLM-8B *during inference* to n tokens (so that it becomes an n-gram model) and use it to generate the sequence. Then, we calculate the fractal parameters $S$ and $H$ and plot those against $n$. As you can see from the figures, $H$ indeed increase as $n$ increases before it converges to some definite value, which makes sense because a larger $H$ indicates more LRD.
> > > >
> > > > If this is the experiment you are looking for, we promise to include a more detailed version of the experiment (using more models and data domains) in the supplementary materials.
> > > >
> > > > Thank you

---

> > > > > ### Comment · Reviewer_UXZb · 2024-08-12
> > > > > **I believe the authors**
> > > > >
> > > > > Thank you so much for your immediate response!
> > > > >
> > > > > >In this experiment, we limit the context length of PaLM-8B during inference to n tokens (so that it becomes an n-gram model) and use it to generate the sequence. Then, we calculate the fractal parameters  and  and plot those against . As you can see from the figures,  indeed increase as  increases before it converges to some definite value, which makes sense because a larger  indicates more LRD.
> > > > >
> > > > > Yes, restricting the context length and varying its size is essentially the same as the n-gram language model. Thus, including such experimental results can be what I expected. Since this conversation is conducted on OpenReview and I believe the authors will keep their promise, I'll raise my score to 7. Note that to see the drastic changes between the successful recent large language models and count-based traditional models, I still recommend you use count-based n-gram language models or PCFGs as your baseline. Of course, it's not mandatory.

---

### Author Rebuttal · Authors · 2024-08-06

We sincerely thank all reviewers for their insightful and constructive feedback. We appreciate the positive feedback on the thoroughness, originality, and importance of our work.

**Presentation**

We will incorporate the reviewers’ suggestions in the revised version of the paper. This includes providing missing references, clarifying the definition of S, and discussing the breadth of tasks covered in our evaluations, which include many tasks such as logical deduction, multi-step arithmetic, disambiguation, translation error detection, and general knowledge all prompted using various strategies (e.g. direct and chain-of-thought with either 0, 3, 5, or 8 shots). We believe our evaluations are quite thorough and diverse, and will clarify this more in the revised version of the paper.

**Public Checkpoints**

Please note that we have also applied the same methodology during the rebuttal period using the released gemma-2b checkpoint (https://huggingface.co/google/gemma-2b), in response to reviewer cohu's comments. Gemma 2B is much smaller than all of the models we have used in our paper. Yet, we generally observe similar conclusions. First, we have a self-similar structure with near perfect linear fits in a log-log plots. Second, we also observe power laws using the rescaled-range analysis, with a Hurst exponent of about 0.7 in most domains except DM Mathematics (smallest Hurst exponent of about 0.58) and GitHub (largest Hurst exponent of about 0.82), in general agreement to the rest of the models. Please see the attached PDF file for the figures generated using Gemma 2B. These results are based on a subset of the Pile validation split due to the short time constraint (for the rebuttal), so we plan to run the analysis on the full validation data and include those results in the supplementary materials of the paper.

**Information-theoretic complexity**

We discuss in Lines 84-91 why we believe this information-theoretic complexity is meaningful for our analysis. It corresponds to an intrinsic, irreducible description of language and the minimum compute overhead to comprehend/decode it. One experimental evidence in psychology for why this characterization is natural for language comes from reading time measurements, which turned out to correlate well with information-theoretic complexity. In addition, fractal behavior has a clear interpretation in this context: e.g. surprising paragraphs will follow predictable paragraphs, in a manner that is statistically similar to how surprising sentences follow predictable sentences. Please see the discussion and references in Lines 84-91. We will clarify this more in the revised version of the paper.

While our work focuses on this specific aspect of language modeling, we recognize the potential for future research to explore more language structures, and we believe our work is a first step in the direction of exploring the relationship between language modeling and fractals. In addition, our findings regarding the self-similarity of "surprise" in language and the connection between Hurst parameter and downstream performance both are robust findings (as we show in the paper) and can be quite valuable on their own right.

**Follow-up**

If we have satisfactorily answered your questions, we hope you would consider revising your scores. Otherwise, please let us know if there are any other questions or concerns, so we can respond to them during the discussion period.

---

### Decision · Program_Chairs · 2024-09-25

**Decision:**

Accept (poster)

**Comment:**

In this paper, the authors investigate the potential presence of fractal structures in language modeling, particularly within the framework of recent models that predict the next token. To support this idea, they focus on key properties of fractal structures: self-similarity, long-range dependence, and information-theoretic complexity. These properties are quantitatively described using metrics commonly associated with fractal phenomena, including the self-similarity exponent, Hurst parameter, fractal dimension, and the Joseph effect. Experimental results indicate that text modeling characteristics from ArXiv, GitHub, and Wikipedia datasets within The Pile's validation split, for sequences longer than 4K tokens, align with the authors' assumptions. Additionally, analyses of downstream tasks such as BBH, MMLU, and GSM8K reveal that task performance can generally be predicted from language modeling performance under these assumptions, except when sequence lengths during training are considered.

Overall, I find the paper to be interesting and the contribution solid. The weaknesses raised by the reviewers do not justify major corrections and were overally properly addressed by the authors. Yet, I do encourage the authors to seriously take their promise to make specific changes to the paper, as some reviewer comments have the potential to improve the paper significantly (e.g. comments on the linguistic properties of the baseline).